# Double and Triple Combinations of Broadly Neutralizing Antibodies Provide Efficient Neutralization of All HIV-1 Strains from the Global Panel

**DOI:** 10.3390/v14091910

**Published:** 2022-08-29

**Authors:** Evgeniya A. Kochina, Felix A. Urusov, Artem A. Kruglov, Dina V. Glazkova, German A. Shipulin, Elena V. Bogoslovskaya

**Affiliations:** 1Centre for Strategic Planning and Management of Biomedical Health Risks, Federal Medical Biological Agency, Moscow 119992, Russia; 2Izmerov Research Institute of Occupational Health, Moscow 105275, Russia

**Keywords:** HIV-1, Env, broadly neutralizing antibodies, N6, PGT128, PGDM1400, VCR07-523, CAP256-VRC26.25, 10-1074, 10E8, DH511.11P, HIV-1 global panel

## Abstract

The use of broadly neutralizing antibodies (bNAbs) is a promising approach to HIV-1 treatment. In this work, we evaluate the neutralizing activity of the following HIV-1 bNAbs: VCR07-523, N6, PGDM1400, CAP256-VRC26.25, 10-1074, PGT128, 10E8, and DH511.11P, which are directed to different Env surface epitopes. We used the global panel of HIV-1 pseudoviruses to analyze the bNAbs’ potency and chose the most potent ones. To achieve maximum neutralization breadth and minimum IC_50_ concentration, the most effective antibodies were tested in double and triple combinations. Among the doubles, the combinations of N6+PGDM1400 and N6+PGT128 with IC_50_ ≤ 0.3 µg/mL proved to be the most effective. The most effective triple combination was N6+PGDM1400+PGT128. Our data demonstrate that this combination neutralizes pseudoviruses of the global HIV-1 panel with IC_50_ ≤ 0.11 µg/mL and IC_80_ ≤ 0.25 µg/mL.

## 1. Introduction

HIV infection continues to be one of the most serious public health problems. The total number of HIV-1-infected people is estimated at 38 million people in 2022 [1]. Today, 28.2 million people receive antiretroviral therapy, but this is not enough to prevent the spread of HIV-1 [1]. Current antiretroviral drugs have a number of disadvantages such as toxicity, high cost, and the necessity for lifelong uptake. Another problem is drug resistance, which develops in some patients [2]. Thus, despite significant progress in the development of drugs against HIV-1, it is necessary to create new approaches to HIV-1 treatment.

For a long time, it was believed that the antibody response does not have a significant impact on the course of infection due to high virus variability. However, in 2008, broadly neutralizing antibodies (bNAbs) that are capable of neutralizing HIV-1 of different subtypes were first discovered [3,4]. The mechanism of bNAb formation during long virus persistence was attributed to the repetitive hypermutation of immunoglobulin loci, which contributed to the evolution of antibodies’ specificity [5]. Several conserved regions of the Env glycoprotein serve as targets for bNAbs (Figure 1).

In recent years, the list of bNAbs has expanded significantly. The new generation of bNAbs is more active and is capable of providing protection against a wide range of viral strains. The antiviral properties of several individual bNAbs have been investigated in clinical trials, which have shown their high potency and ability to reduce viral load in patients [6,7].

To achieve the maximum neutralization breadth and prevent the formation of antibody-resistant strains, it would be advantageous to use bNAb combinations. It is important to note that a combination of antibodies does not always provide an additive effect. Sometimes, conformational changes in the Env protein caused by the Env–antibody interaction can prevent the binding of another antibody that targets a different region [8]. Therefore, it is not possible to predict the functionality of antibody combinations, as each requires careful empirical testing on different viral strains. In this work, we have identified effective double and triple combinations of antibodies with 100% neutralization breadth when tested on a global panel of HIV-1 pseudoviruses.

Based on the literature data, we selected eight effective antibodies that recognize four conservative Env regions (two per each region): (1) CD4 binding site (CD4bs), (2) V1/V2 region, (3) V3 region, and (4) membrane-proximal outer region of gp41 (MPER) (Figure 1). The chosen antibodies have not yet been tested together. In addition, for some of them, the neutralization efficiency data are controversial [9]. Therefore, we conducted a direct assessment of their neutralizing activity in one experiment.

## 2. Materials and Methods

### 2.1. Design of Antibody Constructs

Nucleotide sequences of antibodies were obtained from the NCBI GenBank database or generated based on amino acid sequences taken from the NCBI Protein database (www.ncbi.nlm.nih.gov/protein accessed on 25 April 2022) using the OPTIMIZER service (genomes.urv.es/OPTIMIZER). To ensure the secretion of the antibody light and heavy chains, an optimized human growth hormone signal peptide sequence MATGSRTSLLLAFGLLCLPWLQEGSA was added to each before the start of the variable domain. Nucleotide sequences encoding the signal sequence, variable, and constant domain of the light and heavy chain of each bNAb were cloned separately into an expression vector, as shown in Figure A1.

### 2.2. Cell Lines

The HEK293FT cell line (Invitrogen, Carlsbad, CA, USA) was used to produce broadly neutralizing antibodies and pseudoviral particles. Cells were cultured in DMEM (Gibco, Langley, OK, USA) supplemented with 10% FBS (Gibco). To assess the efficiency of neutralization of the pseudoviral particles, the TZM-bl cell line (NIH AIDS Reagent Program, Manassas, VA, USA) was used. Cells were cultured in cDMEM:DMEM supplemented with 10% FBS and 25 mM HEPES (Gibco, Langley, OK, USA).

### 2.3. Antibody Production

HEK293FT cells were seeded into a culture flask at a rate of 1.2 × 10^5^ cells/cm^2^. The next day, transfection with PEI MAX (Polysciences Inc., Warrington, PA, USA) was performed. The DNA:PEI ratio was 1:3. The ratio of plasmid DNA encoding the light chain to the plasmid encoding the heavy chain was 2:1 [10]. The day after transfection, the growth medium was changed to serum-free OptiMEM (Gibco, Langley, OK, USA). Seventy-two hours after transfection, the antibody-containing medium was collected, filtered through a 0.45 µM pore size mesh, and concentrated 15-fold by Amicon Ultra 15 (50 kDa) (Merck Millipore, Darmstadt, Germany).

### 2.4. ELISA

To measure the concentration of antibodies, an IgG total-ELISA-BEST (Vector-Best, Novosibirsk, Russia) kit was used. The procedure was carried out according to the manufacturer’s instructions.

### 2.5. Western Blot

The samples and PageRuler Plus protein marker (Thermo Fisher Scientific Inc., Waltham, MA, USA) were applied to a 12% SDS-PAGE gel (each sample contained 20 µg of protein). After electrophoresis under reducing conditions, proteins were transferred to a membrane (Immun-Blot LF PVDF membrane, Bio-Rad Laboratories, Inc., Hercules, CA, USA) by wet transfer. The membranes were blocked in PBS containing 5% skimmed milk powder and 0.1% Tween 20 (Sigma-Aldrich, St. Louis, MO, USA) and incubated overnight at 4 °C with polyclonal HRP-conjugated goat antihuman IgG(H+L) antibodies (Jackson ImmunoResearch Europe Ltd., Ely, Cambridgeshire, UK) at a 1:15,000 dilution. Then, the membrane was washed five times in a solution of PBS and 0.1% Tween 20 (Sigma-Aldrich, St. Louis, MO, USA). Protein complexes were detected with Clarity Western ECL Substrate reagent (Bio-Rad Laboratories, Inc., Hercules, CA, USA) according to the manufacturer’s recommendations and photographed using X-ray film (Fujifilm, Tokyo, Japan).

### 2.6. Production of HIV-1 Pseudoviruses

To obtain HIV-1 pseudoviral particles, HEK293FT cells were seeded in a 75 cm^2^ culture flask and cotransfected with a pSG3ΔENV vector (6.9 μg) and a plasmid encoding the corresponding Env protein from the global HIV-1 panel (3.5 μg) [11]. Panel of Global HIV-1 Env Clones (cat# 12670) was obtained through the NIH AIDS Reagent Program, Division of AIDS, NIAID, NIH, from Dr. David Montefiori. PEI MAX (Polysciences Inc., Warrington, PA, USA) was used for transfection. The DNA:PEI ratio was 1:3. The procedure for the transfection and collection of the cell supernatant containing pseudoviral particles is described in [12]. Fifteen milliliters of viral supernatant were added onto Amicon Ultra-15 (100K) (Merck Millipore, Darmstadt, Germany) and passed through the filter by centrifugation until the residual volume reached 1.5 mL. Then, diafiltration with 10 mL of 1× dPBS per 1 Amicon filter was performed. Finally, the solution was spun to a 1.5 mL volume. The concentrated pseudoviral particles were stored in small aliquots at −70 °C.

### 2.7. Titration of HIV-1 Pseudoviruses

Titration of pseudoviruses was carried out on TZM-bl cells, which contain the HIV-sensitive luciferase transgene, according to the protocol described by Sarzotti-Kelsoe [13]. One hundred microliters of cDMEM were added to each well of the 96-well plates. Then, 25 μL of the fivefold serial dilutions of pseudoviral stock were added. Then, a 100 μL suspension containing 10,000 TZM-bl cells in cDMEM with 8 μg/mL polybrene (Sigma-Aldrich, St. Louis, MO, USA) was added to each well. All samples were assayed in triplicate. Twenty-four hours after transduction, the medium was changed to cDMEM. Seventy-two hours after transduction, the luciferase activity was estimated using a Bright-Glow reagent (Promega, Madison, WI, USA) according to the manufacturer’s instructions. The luminescence level was measured on a Tecan Infinite 200 Pro luminometer (Tecan Group Ltd., Männedorf, Switzerland). Then, an RLU (relative luminescence units) versus virus dilution curve was plotted, and the volume required to achieve a luminescence of about 200,000 RLU was determined. This volume was used in the following neutralization assays.

### 2.8. Neutralization Assay

The neutralization reaction was carried out according to the previously described protocol [13]. Briefly, fivefold dilutions of the antibodies were mixed with the previously defined volume of pseudovirus stock in a 96-well plate, and cDMEM was added up to a volume of 125 µL. Antibody concentrations in the resulting mixtures were 6 µg/mL, 1.2 µg/mL, 0.24 µg/mL, and 0.048 µg/mL. The same total amount of antibodies was taken for combinations. For double combinations, the antibody ratio was 1:1; for triple combinations, the ratio was 1:1:1. Samples were incubated for 1.5 h at 37 °C. Then, a 100 μL suspension containing 10,000 TZM-bl cells in cDMEM with 8 μg/mL polybrene (Sigma-Aldrich, St. Louis, MO, USA) was added to each well. Twenty-four hours after transduction, the medium was changed to cDMEM, and 72 h after transduction, RLU was obtained as described above.

### 2.9. Data Processing

The calculation of IC_50_ was carried out in the Excel (Microsoft Corporation, Redmond, WA, USA) program according to the guidance provided by the Montefiori DC laboratory [14]. RLU derived from the analyzed wells were normalized to the RLU of the control samples, which contained the same pseudovirus without bNAb. The obtained value was converted to a percentage of neutralization. Then, the percentage neutralization was plotted against the concentration of antibodies on a logarithmic scale, and a nonlinear regression model was built. IC_50_ values were defined as the number of antibodies needed to achieve a 50% pseudoviral neutralization. The IC_50_ values given in the tables are the mean values obtained from two independent experiments.

## 3. Results

The following antibodies were used in the work: VCR07-523 and N6 (CD4bs), PGDM1400 and CAP256-VRC26.25 (V1/V2), 10-1074 and PGT128 (V3), and 10E8 and DH511.11P (MPER) [15,16,17,18,19,20,21] (Figure 1). Rituximab, which binds to the human CD20 protein, was chosen as a control because it does not neutralize HIV-1 and is characterized by a high level of expression and stability [22].

Antibodies were produced in HEK293FT cells by transient expression. Growth media containing secreted antibodies were collected and concentrated 15-fold by Amicon Ultra (50 kDa). The resulting concentration of antibodies was measured by performing a Human IgG ELISA. The concentrations of produced antibodies ranged from 16 to 171 µg/mL (Figure 2).

Antibody production in the medium was also confirmed by a Western blot with polyclonal antibodies against human IgG. For most antibodies, we detected two bands that corresponded to the expected molecular weights of the heavy and light chains. Unexpectedly, analysis of the CAP256-VRC26.25 antibody showed an additional band with a molecular weight lower than that of the heavy chain. The origin of this band remains unclear. One explanation is the proteolytic cleavage of the heavy chain by specific proteases in producer cells. However, as shown below, this fact did not substantially affect the neutralizing properties of this antibody (Table 1, Figure A2).

According to the Western blot data, all antibodies are expressed in HEK293FT cells and secreted into the culture medium (Figure 3). Furthermore, we evaluated their neutralizing activity.

### 3.1. Neutralizing Activity of Antibodies

To assess the neutralizing activity of the produced antibodies, pseudoviruses from the global panel of HIV-1 were used [11]. The Env glycoprotein variants of the global panel represent the major genetic variants of the virus. The ability of antibodies or their combinations to neutralize pseudoviruses of the global panel reflects their potential use against the majority of viral strains circulating in the world. Thus, the global panel allows us to reduce the number of measurements needed to analyze the antibodies’ breadth.

We carried out a neutralization assay for all eight antibodies against global panel pseudoviruses, and half-maximal inhibitory concentrations (IC_50_ values) for each antibody–pseudovirus pair were calculated (Table 1, Figure A2).

According to the obtained data, the N6 antibody has the widest neutralization breadth among studied bNAbs and neutralizes all strains of the global panel (Table 1). For all other antibodies, at least one pseudovirus that cannot be neutralized was found. For each Env site, we identified the most effective antibodies against each region of Env (N6, PGT128, PGDM1400, and 10E8), which were further analyzed as a part of combinations.

### 3.2. Antibody Combinations

Next, we used combinations of N6, PGT128, PGDM1400, and 10E8 bNAbs to achieve efficient neutralization of the entire global panel.

We took N6 as a base for each of the double combinations. The N6 antibody provided the maximum breadth of neutralization among all tested bNAbs because it neutralized the entire global panel. However, N6 had IC_50_ values within the 0.1 to 1.34 µg/mL range (Table 1). This range has a suboptimal upper bound. We assumed that the combination of antibodies would provide optimal IC_50_ values and reach the desired neutralization breadth. The following double combinations have been tested (Table 2, Table 3, Table 4 and Table 5, Figure 4): N6 + 10E8, N6 + 10-1074, N6 + PGDM1400, and N6 + PGT128. All of them showed better efficiency in comparison with single antibodies and were able to neutralize the whole panel of pseudoviruses. Among them, the combinations N6 + PGDM1400 and N6 + PGT128 showed higher efficiency (the IC_50_ ranges were 0.06–0.3 µg/mL and 0.01–0.3 µg/mL, respectively, Table 2 and Table 4).

Next, we analyzed two triple combinations of antibodies (Table 2, Table 3, Table 4 and Table 5, Figure 4): N6 + 10E8 + 10-1074 and N6 + PGDM1400 + PGT128. As expected, the triple combinations were more effective than the two-antibody mixes. The N6 + PGDM1400 + PGT128 combination neutralized the entire global panel with an IC_50_ range within 0.01–0.11 µg/mL. It slightly outperformed the N6 + 10E8 + 10-1074 combination, which had IC_50_ values in the range of 0.02–0.15 µg/mL. The N6 + PGDM1400 + PGT128 combination demonstrated extremely high potency against four pseudoviruses: BJOX2000, CE0217, X1632, and 25,710 (IC_50_ < 0.01 µg/mL).

Additionally, we evaluated the efficiency of bNAbs and their combinations by IC_80_ calculation (Table 3 and Table 5). As anticipated, applying a more stringent criterion of 80% virus inhibition led to increased IC_80_ values compared with IC_50_. As a result, the IC_80_ range of N6, 10E8, PGDM, N6 + 10E8, and N6 + 10-1074 moved to the “yellow” zone with moderate neutralizing activity (1–5 μg/mL). Both of the triple combinations showed high neutralizing activity—all IC_80_ values remained in the “green” zone (<1 µg/mL).

## 4. Discussion

The search for effective ways to prevent and treat HIV infection is an urgent public health problem. Unfortunately, no effective vaccine against HIV-1 has yet been developed [24]. The high level of Env variability contributes to viral escape from the antibodies produced in the body [25]. The bNAbs that are able to neutralize a wide variety of HIV-1 strains were found only in a small number of patients and mainly at the late stages of infection [26].

A large number of bNAbs have been isolated and characterized [3,20]. Modifications of bNAbs that lead to increased efficiency and extended bNAb serum half-life have been described [27]. The breadth and efficacy of next-generation antibodies open up prospects for their clinical application in passive immunization against HIV infection [28]. Several bNAbs have already been tested in clinical trials, in which clinical effects were shown, such as viral load decline in viremic subjects or increasing the time to rebound during antiviral treatment interruption [6]. At the same time, clinical trials demonstrated that HIV-1 could escape the action of a single antibody, so bNAbs combinations will be required for such therapy [29]. Therefore, the search for an effective bNAbs combination remains relevant.

Based on the literature data, we selected eight bNAbs against four different Env regions and compared their neutralizing activity. All of them showed different activity against pseudoviruses in the global panel (Table 1). In most cases, our results were in agreement with previously published data (Table 6). According to our observations, the N6 antibody has the greatest neutralization breadth, as has also been shown by other authors [16]. For the other antibodies, we found discrepancies in the neutralization efficiency of individual viral strains. For example, early data demonstrated that the PGDM1400 antibody did not neutralize the CH119, BJOX2000, X2278, and 246F3 viruses [30]. However, we found that PGDM1400 successfully neutralized these viruses. A slight decrease in the effectiveness of the CAP256-VRC26.25 antibody compared to the literature data may be explained by its putative incorrect processing in producer cells, which was revealed using Western blot. In some cases, there are contradictions between the findings of different authors (Table 6). The reasons for these are not always clear, and, most likely, differences occur due to the multistage nature of obtaining the final IC_50_ values. It should be noted that the accuracy of the antibodies’ concentrations can be highly variable between different laboratories due to the different ELISA kits used. This can significantly affect the reliability of the IC_50_ values.

To select the optimal bNAbs combination, it is paramount to use antibodies that are directed to different Env epitopes [34]. This strategy may help us to maximize the number of neutralizable virus strains and prevent viral escape from neutralization. In this work, we studied the efficiency of the following combinations consisting of two antibodies: N6 + 10E8, N6 + 10-1074, N6 + PGT128, and N6 + PGDM1400. The antibodies that initially showed the greatest efficacy were included in the combinations (Table 1). We found combinations to be superior to single antibodies and to be able to neutralize all pseudoviruses of the global panel (Table 2, Table 3, Table 4 and Table 5). When choosing the optimal combination, it is important to pay attention not only to the breadth but also to the efficiency, i.e., to the concentration of antibodies needed to neutralize the virus. This is crucial because increased efficiency will allow us to use a smaller amount of antibodies for administration to patients. To further enhance efficiency, we made two triple combinations, N6 + 10E8 + 10-1074 and N6 + PGT128 + PGDM1400, and analyzed them. Among them, the latter was a little more effective, with an IC_50_ range of 0.01–0.11 µg/mL and an IC_80_ range of 0.01–0.25 µg/mL. Literature data show that the combination of three antibodies increases the efficiency of neutralization and may be sufficient to neutralize 99% of viruses in large panels of pseudoviruses [34]. In the future, we plan to perform additional experiments on expanded HIV-1 panels to study the breadth of the chosen combinations more accurately.

## 5. Conclusions

In the present work, we compared the neutralizing activity of eight antibodies directed to four different epitopes: CD4bs, V1/V2, V3, and MPER. We have chosen the most potent bNAbs against each epitope. Based on these bNAbs, we made several combinations and compared their potency with each other. The most effective triple combination, N6 + PGT128 + PGDM1400, completely neutralized the global panel of HIV-1 pseudoviruses, with an IC_50_ range of 0.01–0.11 µg/mL. This combination is a promising candidate for further testing in preclinical and clinical trials.

## Figures and Tables

**Figure 1 viruses-14-01910-f001:**
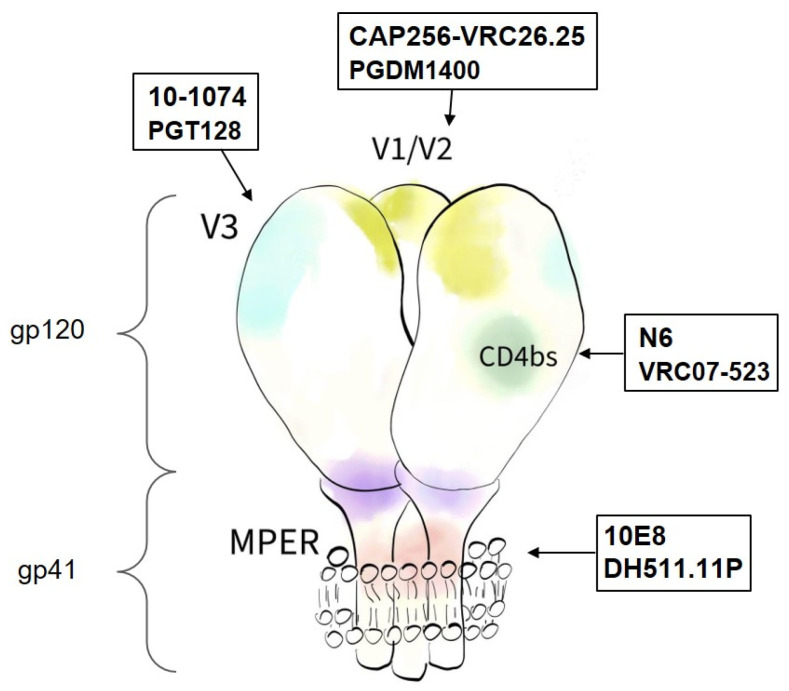
Schematic representation of glycoprotein Env homotrimer on the viral membrane. Each monomer consists of noncovalently bound gp120 and gp41 polyproteins. Epitopes for bNAbs are depicted in different colors. bNAbs used in this work are listed in frames.

**Figure 2 viruses-14-01910-f002:**
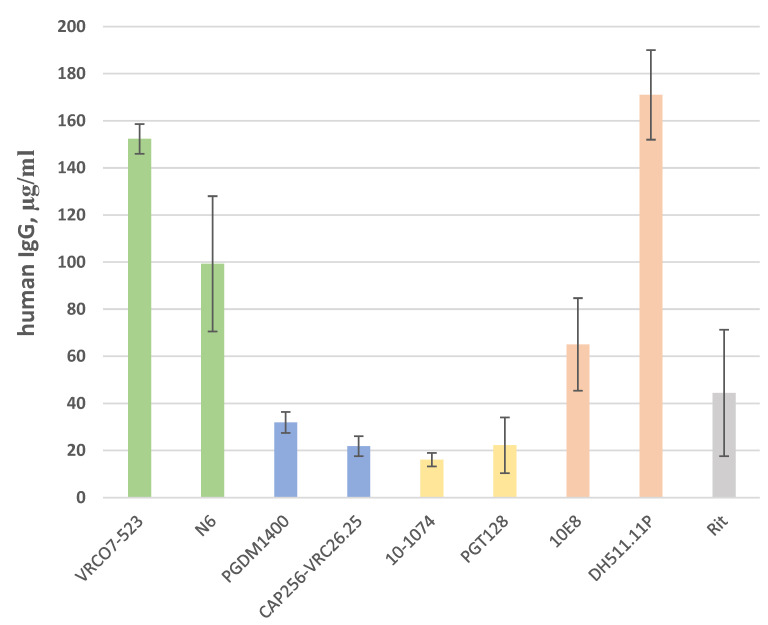
Antibody concentration (ELISA, total human IgG) (n = 3, ±SEM).

**Figure 3 viruses-14-01910-f003:**
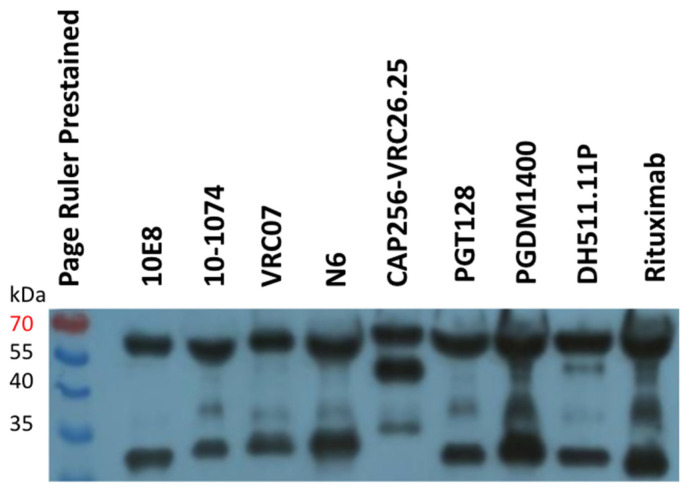
Western blot analysis of broadly neutralizing antibodies under reducing conditions.

**Figure 4 viruses-14-01910-f004:**
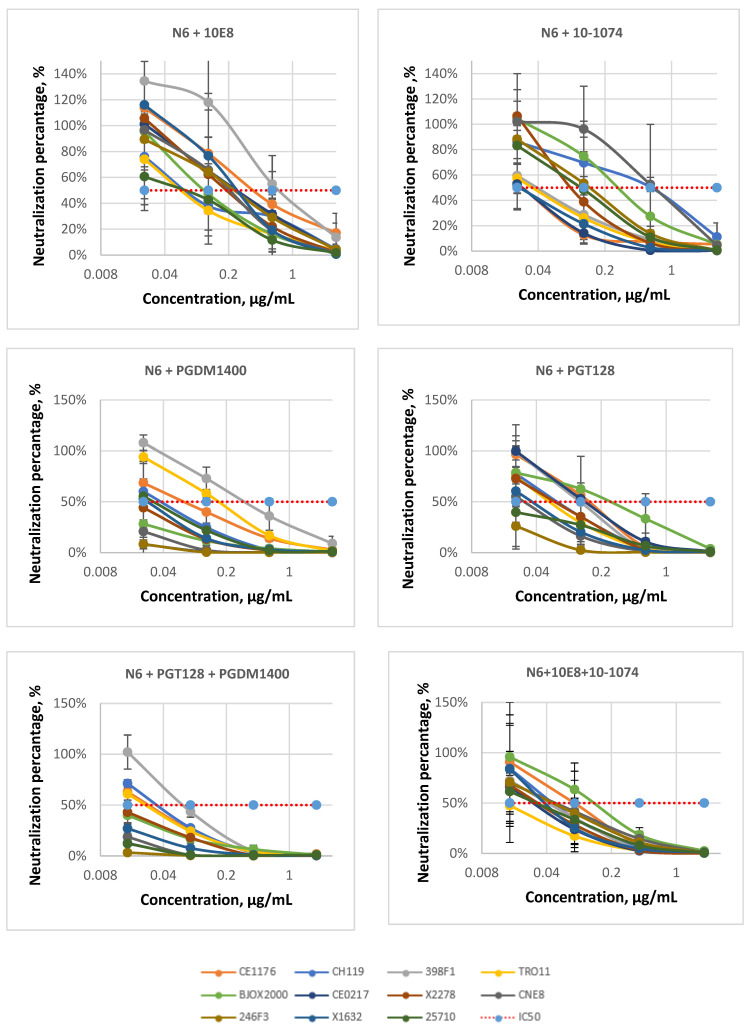
Pseudoviruses neutralization curves for bNAbs combinations (n = 2, ±SEM, on the Y-scale, 100% means no neutralization and 0%—complete neutralization).

**Table 1 viruses-14-01910-t001:** Mean IC_50_ values (n = 2) for single antibodies.

	bNAbs	10E8	DH511.11P	N6	VRC07-523	10-1074	PGT128	CAP256-VRC26.25	PGDM1400
Pseudovirus Strains									
CE1176(C)		0.26	1.52	1.34	2.11	0.21	0.43	>5	0.89
CH119 (CRF07)		0.30	>5	0.21	0.53	0.20	0.33	0.13	0.54
398F1(A)		0.49	2.05	0.87	1.72	0.17	0.32	>5	>5
TRO11 (B)		0.82	0.67	0.70	0.60	0.11	0.08	>5	2.11
BJOX2000 (CRF07)		0.39	3.52	0.31	0.27	>5	>5	<0.01	<0.01
CE0217 (C)		>5	1.02	0.14	0.43	<0.01	0.21	<0.01	<0.01
X2278 (B)		0.36	2.39	0.56	0.56	0.11	0.07	0.01	0.01
CNE8 (CRF01)		0.19	0.99	0.85	>5	>5	0.03	2.24	<0.01
246F3 (AC)		0.46	2.04	0.16	1.24	>5	0.05	0.18	<0.01
X1632 (G)		0.36	>5	0.42	>5	0.07	0.23	<0.01	0.01
25710 (C)		0.03	0.46	0.72	1.47	0.30	0.11	<0.01	<0.01

According to Webb et al., IC_50_ values up to 1 µg/mL mean that the antibody effectively neutralizes the pseudovirus. Antibodies with an IC_50_ in the range of 1 to 5 μg/mL have moderate potency, while IC_50s_ of 5 μg/mL and above have no or very low neutralizing activity [23]. Green: <1 µg/mL; Yellow: 1–5 µg/mL; Red: >5 µg/mL

**Table 2 viruses-14-01910-t002:** IC_50_ values for antibodies N6, 10E8, 10-1074, and their combinations.

	bNAbs	N6	10-1074	10E8	N6 + 10E8	N6 + 10-1074	N6 + 10-1074 + 10E8
Pseudovirus Strains							
CE1176 (C)		1.34	0.21	0.26	0.70	0.03	0.14
CH119 (CRF07)		0.21	0.2	0.3	0.12	0.51	0.09
398F1 (A)		0.87	0.17	0.49	0.84	0.04	0.05
TRO11 (B)		0.7	0.11	0.82	0.15	0.04	0.02
BJOX2000 (CRF07)		0.31	>5	0.39	0.71	0.30	0.15
CE0217 (C)		0.14	<0.01	>5	0.33	0.02	0.03
X2278 (B)		0.56	0.11	0.36	0.24	0.18	0.04
CNE8 (CRF01)		0.85	>5	0.19	0.27	0.48	0.07
246F3 (AC)		0.16	>5	0.46	0.29	0.16	0.05
X1632 (G)		0.42	0.07	0.36	0.33	0.03	0.08
25710 (C)		0.72	0.3	0.03	0.11	0.13	0.05
IC_50_ range		0.14–1.34	0.01–5	0.03–5	0.11–0.84	0.02–0.51	0.02–0.15

Green: <1 µg/mL; Yellow: 1–5 µg/mL; Red: >5 µg/mL.

**Table 3 viruses-14-01910-t003:** IC_80_ values for antibodies N6, 10E8, 10-1074, and their combinations.

	bNAbs	N6	10-1074	10E8	N6 + 10E8	N6 + 10-1074	N6 + 10-1074 + 10E8
Pseudovirus Strains							
CE1176 (C)		3.07	0.57	1.33	2.13	0.29	0.36
CH119 (CRF07)		1.07	0.5	1.33	0.87	1.80	0.29
398F1 (A)		1.98	0.67	1.86	1.73	0.24	0.16
TRO11 (B)		2.5	0.44	2.61	0.64	0.34	0.14
BJOX2000 (CRF07)		0.8	>5	>5	2.63	1.23	0.61
CE0217 (C)		0.9	0.09	>5	1.18	0.21	0.22
X2278 (B)		1.67	2.69	1.10	0.99	0.55	0.24
CNE8 (CRF01)		>5	>5	1.24	0.83	1.75	0.39
246F3 (AC)		0.78	>5	2.18	1.22	0.72	0.36
X1632 (G)		1.62	0.31	1.58	0.87	0.26	0.23
25710 (C)		2.23	1.18	0.34	0.54	0.63	0.28
IC_80_ range		0.78–5	0.09–5	0.34–5	0.54–2.63	0.21–1.80	0.14–0.61

Green: <1 µg/mL; Yellow: 1–5 µg/mL; Red: >5 µg/mL.

**Table 4 viruses-14-01910-t004:** IC_50_ values for antibodies N6, PGT128, PGDM1400, and their combinations.

	bNAbs	N6	PGT128	PGDM1400	N6 + PGT128	N6 + PGDM1400	N6 + PGDM1400 + PGT128
Pseudovirus Strains							
CE1176 (C)		1.34	0.43	0.89	0.18	0.05	0.03
CH119 (CRF07)		0.21	0.33	0.54	0.09	0.03	0.04
398F1 (A)		0.87	0.32	>5	0.16	0.30	0.11
TRO11 (B)		0.7	0.08	2.11	0.14	0.21	0.03
BJOX2000 (CRF07)		0.31	>5	<0.01	0.30	0.12	<0.01
CE0217 (C)		0.14	0.21	<0.01	0.19	<0.01	<0.01
X2278 (B)		0.56	0.07	0.01	0.08	0.01	0.01
CNE8 (CRF01)		0.85	0.03	<0.01	0.18	0.19	<0.01
246F3 (AC)		0.16	0.05	<0.01	0.18	0.16	<0.01
X1632 (G)		0.42	0.23	0.01	0.06	0.04	<0.01
25710 (C)		0.72	0.11	<0.01	0.06	0.06	<0.01
IC_50_ range		0.14–1.34	0.03–5	0.01–5	0.06–0.3	0.01–0.3	0.01–0.11

Green: <1 µg/mL; Yellow: 1–5 µg/mL; Red: >5 µg/mL.

**Table 5 viruses-14-01910-t005:** IC_80_ values for antibodies N6, PGT128, PGDM1400, and their combinations.

	bNAbs	N6	PGT128	PGDM1400	N6 + PGT128	N6 + PGDM1400	N6 + PGDM1400 + PGT128
Pseudovirus Strains							
CE1176 (C)		3.07	0.65	3.25	0.61	0.37	0.23
CH119 (CRF07)		1.07	0.75	1.46	0.45	0.20	0.25
398F1 (A)		1.98	0.75	>5	0.36	0.50	0.24
TRO11 (B)		2.5	0.28	3.80	0.56	0.69	0.22
BJOX2000 (CRF07)		0.8	>5	0.30	1.17	0.30	0.12
CE0217 (C)		0.9	1.00	<0.01	0.70	<0.01	<0.01
X2278 (B)		1.67	0.44	0.36	0.42	0.11	0.12
CNE8 (CRF01)		>5	0.32	0.01	0.53	0.57	0.01
246F3 (AC)		0.78	0.21	<0.01	0.67	0.59	<0.01
X1632 (G)		1.62	0.69	1.14	0.27	0.14	0.04
25710 (C)		2.23	0.50	<0.01	0.34	0.19	<0.01
IC_80_ range		0.78–5	0.21–5	0.01–5	0.27–1.17	0.01–0.69	0.01–0.25

Green: <1 µg/mL; Yellow: 1–5 µg/mL; Red: >5 µg/mL.

**Table 6 viruses-14-01910-t006:** Comparison of the IC_50_ values for studied bNAbs with literature data [21,30,31,32,33].

	Pseudoviruses	CE1176	CH119	398F1	TRO11	BJOX2000	CE0217	X2278	CNE8	246F3	X1632	25710	
bNAbs													
**10E8**													[30]
				nd	nd						nd		[32]
		nd		nd			nd	nd		nd		nd	[21]
													**Our data**
**DH511.11P**		nd		nd			nd	nd		nd		nd	[21]
													**Our data**
**N6**													[31]
													**Our data**
**VRC07-523**			nd			nd			nd	nd	nd		[30]
													[31]
													**Our data**
**10-1074**													[30]
			nd	nd		nd	nd	nd	nd	nd	nd	nd	[33]
													**Our data**
**PGT128**			nd	nd		nd	nd	nd	nd	nd	nd	nd	[32]
													[30]
													**Our data**
**CAP256-VRC26.25**													[30]
													**Our data**
**PGDM1400**													[30]
													**Our data**

Green: <1 µg/mL; Yellow: 1–5 µg/mL; Red: >5 µg/mL.

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
