# Peer review of "Double and Triple Combinations of Broadly Neutralizing Antibodies Provide Efficient Neutralization of All HIV-1 Strains from the Global Panel"

_viruses, 2022, doi:10.3390/v14091910_

Round 1

Reviewer 1 Report

Kochina st al., reported highly effective combinations of broadly neutralizing antibodies (bNAbs) against HIV in terms of neutralization using the global panel of HIV-1 pseudoviruses.

HIV-1 bNAbs have received significant attention given their potential as therapeutic tools for HIV-1 treatment and prevention. A number of bNAbs have successfully been tested in humans, and modified versions and bNAb combinations are now being evaluated. It has become clear that monoclonal antibody therapeutics need to have exquisite potency and breadth, ideally targeting multiple trimer epitopes simultaneously in order to avoid the selection of neutralization resistant viral variants. And it remains to be determined what the ideal combination of epitope targets and/or the minimum required number of epitope specificities is to avoid viral escape. From this point of view, the report will take a significant attention from broad readers in this field. However, optimal combinations of bNAbs has been assessed using a larger panels of viruses including a panel of clinical isolates. Because the viruses in vivo consisted of quasispecies the best combination have to be determined based on statistical model such as the “Bliss Hill model (BH model)” in the previous work reported by Wagh, et al., PLoS Pathog., 2016 using data from larger sets of pseudovirus panel.

 Major point:

It is pf note that the finding in this report that N6, which is a best in class antibodies consisted of the best combination has not reported previously. However, the global panel of 11 viruses may not enough to conclude the combination can be the best comparing previous work. Also, evaluation of the effect of the combination by IC80 or IC90 in addition to IC50 with some statistical model have to be done.

Minor point:

line 101: Western blot analysis

I suppose the authors used anti-human IgG conjugated with HRP in the western blot assay. The method have to be accurately written.

In Fig.3, two different VHs were observed for CAP256-VRC26.25. The results have to be discussed in detail.

I think the neutralization curve shown in Fig. 4 and Appendix should be in the style of percent inhibition curve with x-axis in log-scale.

All the description of IC50 have to be IC50 (subscript).

Reviewer 2 Report

In this manuscript, Elena V. Bogoslovskaya and her colleagues chose eight reported HIV-1 neutralizing antibodies and evaluated their neutralization activity under single, double and triple antibody combinations. Firstly, they produced and quantified antibodies. Then, with these antibodies, they tested their neutralizing activity against a panel of HIV-1 pseudoviruses and found that N6 + PGT128 + PGDM1400 combination was the most potent one, completely neutralizing the global panel of HIV-1 pseudoviruses with an IC50 range of 0.01–0.11 μg/ml. This study's experimental design is reasonable, and the data is well presented. The discussion is also adequate. Therefore, I only have a few comments below.

  1. Could you add figure 1C to show each antibody's targeting site in the Env sequence? Although both VRC07-523 and N6 target the CD4 binding site region, the neutralization activity was distinct, according to table 1. It would be easier for readers to compare and understand the results if you showed the specific targeting site.
  2. What is the logic to showing your bNAbs in table 1? It would be better to organize them based on the CD4bs, V1/V2, V3, and MPER, which is the same order shown in figure 1b.

Round 2

Reviewer 1 Report

No further comments.